# Vaccination Coverage during Childhood and Adolescence among Undergraduate Health Science Students in Greece

**DOI:** 10.3390/children9101553

**Published:** 2022-10-13

**Authors:** Elisabeth K. Andrie, Evanthia Sakellari, Anastasia Barbouni, Artemis K. Tsitsika, Areti Lagiou

**Affiliations:** 1Department of Public and Community Health, University of West Attica (UNIWA), 196 Alexandras Avenue, 11521 Athens, Greece; 2MSc Program “Str. of Developmental & Adolescent Health”, National and Kapodistrian University of Athens, Mesogeion 15-Goudi, 11527 Athens, Greece

**Keywords:** immunization coverage, vaccine-preventable diseases, young adults, healthcare services

## Abstract

High rates of vaccination coverage in childhood represent one of the most important cost-effective tools of primary prevention and have substantially reduced the incidence of and mortality from vaccine-preventable diseases globally. Vaccination coverage of young people has not been well estimated in Greece. Anonymous self-completed questionnaires and the participants’ Child Health Booklet were used to estimate complete vaccination coverage of mandatory vaccinations against vaccine-preventable diseases of undergraduate students at the University of West Attica during the academic year 2020–2021. Student’s t-tests were used to estimate mean values. Overall, 79% (95% CI: 78–81%) of study participants were fully vaccinated. This coverage was lower in males especially for vaccines that should be done during childhood (*p* = 0.045). High vaccination percentage (>90%) was observed for the meningococcus group A, C, W135, Y, measles-mumps-rubella, hepatitis B and meningitis C vaccine. Nevertheless, suboptimal coverage was assessed for the booster dose of tetanus, diphtheria, pertussis, for the human papillomavirus vaccine among girls, for the tuberculosis, for the meningococcus group B, for the pneumococcal, and for the seasonal influenza vaccines. In conclusion, the promotion of routine vaccination programs for young adults needs to be strengthened. An identification system for under-vaccinated students, an efficient reminder system and university campus vaccine program practices should be established, particularly among males.

## 1. Introduction

Vaccines are estimated to be among the most important cost-effective public health interventions worldwide and have resulted in the impressive decline and regional elimination of illness, disability, and death caused by vaccine-preventable diseases (VPDs) [1,2,3].

Assessing the vaccination coverage has been essential to the success of immunization programs, in order to measure the vaccination health services, to enhance the service to underserved groups, and to evaluate the success of immunization in public health improvement [4]. Additionally, achieving a high vaccination coverage is an essential component of international, national, or local public health goals, applied by the members of the World Health Assembly [5]. Routine vaccination data or periodic evaluation of vaccination coverage studies have been proposed by the WHO to assess the expansion of national immunization services [6,7,8]. The European Centre for Disease Prevention and Control (ECDC) has founded a network to collect VPDs vaccination data of national vaccination programs in the European Union [9,10].

Although vaccines are highly accepted by the general population, even in countries with high vaccination compliance, a significant number of children, adolescents, and young adults are not completely vaccinated due to a lack of opportunities or various concerns and misperceptions [11,12,13,14]. Emerging adulthood, a distinct developmental period from the late teens to the late twenties with unique changes in physical, emotional, and psychological development, may have a notable impact on immunization compliance with national schedule. Although, during childhood, the compliance to preventive methods entirely depends on parents’ and pediatricians’ suggestions, during adolescence and young adulthood it is the result of the information received mainly via their peers and the social media [15].

Immunization against VPDs evoke high immune protection during childhood but progressively weakens in adolescence and young adulthood; therefore, a higher-than-expected incidence of VPDs is observed in young people [15]. Additionally, higher circulation of pathogens among teens and emerging adults, evokes a high incidence of VPDs infections among incompletely vaccinated younger children or susceptible adults. Emerging adults, who are under or unimmunized during childhood, enter in higher education without updated vaccination status assessment [16] Consequently, they evoke a high risk of disease susceptibility [17] as they spend their time in close settings contributing to the possibility of serious infectious disease outbreaks. Several VPDs can cause complications more often in adults than children of lower education level, while infection of young adults is associated with increased risk for serious morbidity. University students and especially health science students during their education have increased risk of acquiring and transmitting VPDs to other patients. In 2013, the Greek Ministry of Health recommended the immunization schedule for healthcare workers (HCWs) against VPDs. Nevertheless, there is no central registry or identification system documenting vaccination coverage for HCWs and health science students in Greece. 

Although Greece has adopted one of the most updated immunization programs in Europe, like other European countries, it has not established a formal and continuous registry to assess vaccine compliance rates so far [11]. Estimations of the real condition epidemiological data are provided by limited studies with sample and methodological limitations. To our knowledge, published data addressing the evaluation of vaccination rates of all mandatory vaccinations against VPDs according to the Greek National Immunization Program (NIP) of Greek emerging adults are scarce.

This study aimed: (a) to assess complete immunization coverage of mandatory vaccinations according to the Greek NIP, of undergraduate health science students attending one of the largest Universities of Greece, the University of West Attica; (b) to evaluate the vaccination coverage of health science students at the University of West Attica against VPDs recommended by the Greek Ministry of Health for HCWs. Our study findings could help academic health services in creating programs to promote increased vaccination coverage among undergraduate students. In addition, our results may also guide the establishment of program practices geared towards catching-up not sufficiently vaccinated academic students, with special interest to health care students.

## 2. Methods

Our cross-sectional study was conducted at the University of West Attica during the academic year 2020–2021. Health science undergraduate students, aged 18–30 years old, were recruited using the convenience sampling method. The survey was sent to health sciences students who have been educated on the topic “Vaccinations against VPDs according to the Greek NIP”. Following the class, the students had the opportunity to have their Child Vaccination Booklet checked by their tutor (first author) according to the current recommendations for standard vaccinations. The inclusion criteria for the study were students (males and females) aged 18–30 years old.

Invitation emails including the study information, explaining the importance of participation, the names of the researchers, and a link to complete the survey, were sent to participating students. This email also informed the subjects that by clicking the link for the survey completion, they were giving their informed consent simultaneously. The survey was filled out anonymously on a voluntary basis. Participants were asked to use their Child Health Booklet for vaccine documentation and history of VPD natural infection.

The following data were collected: demographic characteristics (age, sex) and vaccination history (including number of doses) against measles, mumps, rubella, varicella, hepatitis A, hepatitis B, tetanus–diphtheria, pertussis, poliomyelitis, meningococcus group B, meningococcus group C, meningococcus groups A, C, W135, Y, tuberculosis (BCG), pneumococcal disease (PCV), human papillomavirus (HPV), and seasonal influenza.

By the age of 10 years, complete vaccination was defined according to the 2020 modification of Greek NIP. More specifically: (1) two doses for measles, mumps, rubella (MMR), (2) two doses for varicella (Var), (3) two doses for hepatitis A (HAV), (4) three doses for hepatitis B (HBV), (5) five doses for diphtheria-tetanus-acellular pertussis (DTaP, TdaP), (6) four doses for inactivated polio vaccine (IPV), (7) four doses for Hemophilus influenzae (Hib), (8) ≥ three doses for pneumococcal (PCV), (9) ≥ one dose for meningococcus serogroup C vaccine (MCC), (12) one dose for Bacillus Calmette–Guerin vaccine (BCG), and (13) ≥ two doses for meningococcus group B vaccine (4CMenB). 

For adolescence (>10 years), complete vaccination was defined according to the 2020 NIP. More specifically: (1) ≥ two doses for human papillomavirus vaccine bivalent/quadrivalent or nine-valent (HPV) for girls, (2) the adolescent booster dose: sixth dose for Tetanus, diphtheria, acellular pertussis (Td or Tdap), and (3) one dose for meningococcal conjugate group A, C, W135, Y vaccine (MenACYW). Complete immunization for influenza was accepted as one dose of influenza vaccine in the past season. 

According to the Greek Ministry of Health recommendations for HCWs, complete vaccination for health science students was defined as follows: two vaccine shots for measles, mumps, rubella (MMR), two shots for varicella, two shots for hepatitis A and three shots for hepatitis B, one booster dose for Td or Tdap or Tdap-IPV during the last 10 years, and one dose of influenza vaccine in the past season.

For each participant, the vaccines that he/she had fully done, following the scientists’ recommendations, were summed and this sum was divided by the number of vaccines that they should have fully done and then multiplied by 100. Thus, a vaccination percentage was computed for each participant, and it was further analyzed.

The study was approved by the Ethics Committee of the University of West Attica (approval code: 67281, date: 6 September 2021.

## 3. Statistical Analysis

Quantitative variables were expressed as mean values (SD), while qualitative variables were expressed as absolute and relative frequencies. For the comparison of proportions, chi-square and Fisher’s exact tests were used. Student’s t-tests were computed for the comparison of mean values. Bonferroni correction was used in order to control for type I error. Paired Student’s t-tests were used for comparison between the vaccination percentage for vaccines that should be done during childhood and the one for those that should be done during adolescence. All reported p values are two-tailed. Statistical significance was set at *p* < 0.05, and analyses were conducted using SPSS statistical software (version 24.0).

## 4. Results

Table 1 shows the sample characteristics. A total of 501 health science students completed the questionnaire (response rate of 32.3%) 15 respondents were dropped because they were over 30 years old. Therefore, the sample consisted by 486 students, aged from 18–30 years old, with mean age 22.2 years (SD = 1.4 years). Most students were females, with the percentage being 85.8%. Almost half of the students (50.2%) were 18–22 years old. 

Table 2 shows full vaccination coverage in total sample and comparison according to age and gender. The full vaccination coverage of mandatory vaccinations against VPDs according to the Greek NIP was 79% (95% CI: 78–81%). This percentage tended to be lower in males, and it did not differ significantly between older (23–30 years old) and younger (23–30 years old) students. The full vaccination coverage for vaccines that should be done during childhood and those that should be done during adolescence are presented in Table 3. The coverage for the vaccines in childhood was 79% (95% CI: 78–80%) and it was similar to the one for the vaccines during adolescence (78%, 95%CI: 75–81%, *p* = 0.452) (Figure 1). The full vaccination coverage for vaccines that should be done during childhood was significantly lower in males (*p* = 0.022). Figure 2 shows a plot describing full vaccination coverage in total and for vaccines that should be done during childhood and adolescence according to the 2020 NIP. The error bars represent percentages along with 95% Confidence Interval.

The percentages of being appropriately vaccinated in each vaccine according to the 2020 NIP are presented in Table 4, in total sample and by gender. The highest percentage in being appropriately vaccinated was found for MenACYW vaccine (100%), followed by MMR (96.1%) and HBV (95.1%). High coverage was observed for childhood immunizations: meningitidis C (MenC; 93.1%), varicella (Var; 86.3%), poliomyelitis (83.3%), and hepatitis A (HAV; 83%). Nevertheless, lower percentages were observed for the booster dose of tetanus, diphtheria, pertussis (70.4%), for the human papillomavirus vaccine (HPV 75.6%) among girls, for the tuberculosis (BCG; 73.8%), for the meningococcus group B (Men B; 17.6%), for the pneumococcal (PCV; 13.6%), and the seasonal influenza vaccines (29.4%). The percentages of being vaccinated against HBV, MMR, and Var were significantly higher in females. Regarding the rest of the vaccines, there were no significant differences between males and females. 

Table 5 shows full vaccination coverage of participants for health professionals (%) in total sample and by their characteristics (age and gender). Full vaccination coverage for health professionals according to Greek Ministry of Health recommendations was 84% (95% CI: 82–85%). There was no significant difference between males and females or between the two age groups. There was no significant difference of this percentage among all departments (data not shown).

## 5. Discussion

Our study indicated the vaccination coverage of mandatory vaccinations against VPDs according to the Greek NIP, of undergraduate health science students attending the University of West Attica. Our findings suggest a suboptimal full vaccination coverage of our sample, as less than 80% of the participant students declared that they had received all recommended vaccines against VPDs. Additionally, there is evidence that one out of five of undergraduate health science students have not been completely vaccinated according to the NIP schedule. The complete immunization percentage is lower than the WHO’s goals (90%), [5], but higher than previous research (61.5%) performed in a major district of central continental Greece [16]. International studies conducted mainly among health science students showed that their vaccination status is partly inadequate [18,19,20,21,22,23,24].

In Greece, vaccination recommendations are included in the NIP which is periodically updated by the National Vaccination Committee. The last modification was applied in 2022. Vaccines, recommended by the NIP, are administered free of charge, while those not yet included in the NIP are financed by recipients. Vaccines are administered by public child health care services and private physicians. The Greek NIP has been modified several times since 1990 (our cohorts’ lowest birth range), as presented in Table 6. 

Our findings suggest adequate vaccination coverage for certain vaccines. Namely, for HBV (95.1%), MMR (96.1%), MCC (93.1%), and MenACWY (100%), the percentage was higher compared to previous studies in Greece, conducted mainly for health science students. This study expands on prior research indicating an increasing compliance over the years [18,25,26,27,28]. Nevertheless, further research is needed, especially regarding certain vaccines.

In our study, full vaccination coverage of mandatory vaccinations tended to be lower in males. Prior research indicates that gender differences can be attributed to differences in knowledge or risk perception about VPDs and vaccines and/or differences in compliance rates with vaccination recommendations [28]. To our knowledge, gender differences regarding immunization coverage have not been previously suggested. Further studies should be conducted to investigate this gender effect.

We found noticeable gaps for the participants’ booster dose of Td/Tdap. For Td/Tdap, approximately 70% of our study subjects are immunized, suggesting that the Td component uptake remains far below the 90% WHO goals [5]. These findings are supported by another study conducted in 2021 among dental students [27] indicating that 63.2% of dental students had received a booster shot against tetanus–diphtheria. During the last decade, completed vaccination coverage, concerning the Td/Tdap vaccine of Greek healthcare students, ranged from 74.6% to 80.2% [18,25]. In 1951, the whole-cell vaccine has been introduced in the Greek NIP and the acellular type in 1997. Doctors of the private sector used booster vaccines at the beginning, before they were included in the Greek NIP, in 2008. According to the Greek NIP, children are vaccinated with a diphtheria, tetanus, and pertussis (DTaP) vaccine at the age of 2, 4, and 6 months as well as with booster doses at the age of 15–18 months and 4–6 years [29]. DTaP is included in four childhood combination vaccines that include other vaccines (e.g., IPV, Hib, HepB). Since 2008, Tdap vaccines are recommended for adolescents. The present study findings indicated that the recent recommendation for Tdap immunization during adolescence (at 11–12 years) has not been widely adopted in Greece. Adolescents and young adults are susceptible to the disease since the pertussis vaccination immunity lasts approximately 5–8 years after the last booster dose [7,30]. Nowadays, pertussis incidence of adolescents and adults is higher, diverting them to reservoirs of disease transmission to infants. As the coverage levels for the tetanus–diphtheria vaccine is one of the best indicators of health-system performance, we assume that the coverage of Td/Tdap vaccine has seriously declined during the financial crisis (2009–2015) when thousands of Greeks were left without social security coverage [31].

It is noteworthy in our study that 75.6% of participants reported that they had been vaccinated against HPV (two or three doses of vaccine) indicating an increasing trend. Nevertheless, compliance with the HPV vaccine among female higher education students in Greece remains suboptimal. In 2008, the National Vaccination Committee introduced the HPV vaccine was in the Greek NIP. The recommendation has been so far, to vaccinate females between the age of 12–15 years, and for the age 16–26 years as a catch-up [32,33]. The vaccine is provided free of charge. The vaccination is provided free of charge by public health care services and private physicians.

Previous research investigating the immunization coverage of HPV vaccine among the young females in Greece is limited. Previous studies concerning Greek university health science students have reported relatively wide fluctuating vaccination coverage rates from 10.47% to 44.3%, while concerning the public, a respective rate of 11% has been reported [32,33,34,35,36]. Across European countries the coverage rates range between 30% and 80% [9]. Specifically, among college students, vaccination coverage rates range from 12% in the UK to 67% in Germany [37,38]. Our study indicates that the vaccination uptake percentage increased gradually over the last decade. Nevertheless, compliance with the HPV vaccine among female higher education students in Greece remains suboptimal, even though vaccination is free of charge. Although it has been introduced to NIP over 14 years ago, knowledge deficiency, various concerns and misperceptions have been proved a critical barrier to vaccination [39]. Health education and promotion initiatives should be developed aiming at adolescents and young adults, since early HPV vaccination is critical for improving the HPV immunization rate [40].

Hepatitis A vaccine (HAV) was included in the Greek NIP in 2007. According to our findings, 83% of participants had received two doses of HAV. Considering the fact that the disease is prevalent in Greece and youths comprise a high risk group for hepatitis A, suboptimal rate highlights the need for coverage increase [2,39,41]. Regarding varicella, consistent with previous national reports, 43.3% of students had reported history of the disease [42]. In our study, about 17% of the sample who reported no varicella history were unvaccinated adolescents and single dose recipients and therefore are susceptible to the disease [43]. Young adults’ sufficient immunization can prevent transmission and considerable morbidity in older age groups and reduce the risk of congenital varicella syndrome as well [44]. 

A notable finding of our study was the suboptimal coverage for the meningococcal serogroup B (MenB) vaccine (17.6%). The MenB is recommended only for children with asplenia or complement deficiency due to the low incidence of the disease in Greece and the high cost of the vaccine.

In contrast, the MenC vaccine showed a notable coverage increase (93.1%) compared to previous national research of 2006. In Greece, the MenC vaccine has been administered since 2001 and has been recommended in NIP since 2005 for older children and adolescents. Previous research indicated vaccination coverage of the MenC vaccine from 20.7% (2001) to 51.4% (2005) [45]. Since April 2011, the MenACYW vaccine has been recommended by the NIP as a booster dose for adolescents at the age of 11–16 years. 

As for the pneumococcal vaccine (PCV), the low coverage (13.6%) that we observed in our study is alarming. The PVC vaccine was included in the NIP in 2006 with the recommendation of three doses for PCV (two shots in the first year of life and the third in the second year) and one shot over the fifth year. The suboptimal immunization rate for PVC vaccine highlights the need for higher coverage among young adults, since the disease is still prevalent in Greece [46].

Similar to previous reports [45], our study vaccination coverage of influenza in the last year was 29.4%. We also found that 64.1% of participants had received at least one shot in the past. Our findings are consistent with previous research indicating a low annual vaccination rate among medical students in Spain [47,48]. In addition, a study conducted among medical students at the Frankfurt University hospital in Germany reported low intention rates of receiving the influenza vaccine [49]. Given the fact that vaccinations are administered free of charge, it seems that their unsatisfactory immunization rates may be attributed to an efficient vaccine reminder and delivery system targeting this age group.

To our knowledge, international data concerning vaccination rates of mandatory vaccinations for health science students are scarce, referring mainly to medical and dental students [19,20,50,51]. Previous research suggests that health science students are not adequately informed about their vaccination coverage. In our study, the majority of participating students completed mandatory vaccinations against VPDs according to the recommendations of the Greek Ministry of Health for HCWs. Our findings are in accordance with the limited similar studies conducted in Greece [25,27]. However, it should be noted that 16% of health science students were not fully vaccinated. Suboptimal coverage among health science students indicates the relevant age-specific vaccination coverage in the general population and probably the lack of awareness and direct access to vaccination services of health science students as well [22,25]. Nevertheless, health science students during their clinical education are at high risk of acquiring VPDs and transmitting them to their patients. 

## 6. Limitations

Our study was conducted in one of the largest universities in Greece and aimed to study a sizable number of students of all academic years. In addition, it is one of the limited studies evaluating complete immunization coverage of undergraduate students conducted in Greece during the last decade.

The novelty of the present study is the fact that it is one of the limited studies conducted to provide vaccination coverage of all vaccines recommended by the Greek NIP addressing emerging adults. To the best of our knowledge, there are no recent studies in Greece examining simultaneously the full immunization coverage against VPDs among undergraduate students.

Our results are based on self-report answers, transcribed from the official health booklet of students who have been recently educated on the topic “vaccinations against vaccine-preventable diseases according to the NIP”; therefore, we believe information bias is restricted. Furthermore, due to the anonymity of self-reporting data, selection and reporting biases are substantially limited.

The response rate of 32.3% appears in line with previous findings (15% to 36.4%) of other similar studies conducted in Greece during the last decade, particularly addressing health science students.

Although the study was conducted in an academic institution in a large metropolitan area, our results may represent the whole Greek population, because there are no major differences in socio-demographic characteristics between the urban population in other Greek cities (which accounts for nearly 80% of the total country population).

Additionally, findings from this cross-sectional study have not revealed associations between vaccination coverage and other sociodemographic variables. An implication for further research could be to explore young adults’ motives and barriers regarding students’ vaccination.

## 7. Conclusions

Immunization against VPDs is essential for young people in closed settings and especially for those in university campuses. A national immunization registry to access the vaccination coverage rates of young adults should be established in Greece.

Our results indicated suboptimal immunization coverage among our sample’s undergraduate students, indicating that the promotion of routine vaccination programs for young adults needs to be strengthened. An identification system for under- or non-vaccinated students should be developed as well as an efficient reminder system and university campus vaccine program practices geared toward catching-up. Strategies to access this age group, with an emphasis on health science students, should be further investigated.

## Figures and Tables

**Figure 1 children-09-01553-f001:**
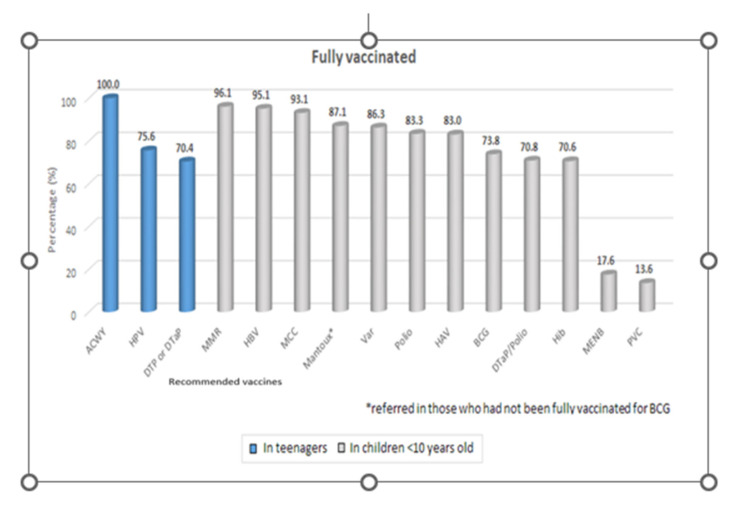
Percentages of being appropriately vaccinated in each vaccine.

**Figure 2 children-09-01553-f002:**
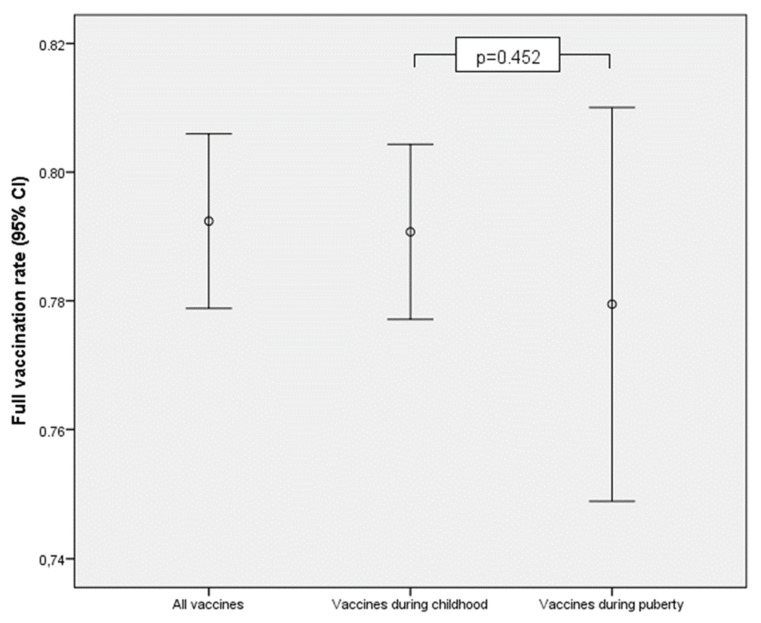
Full vaccination percentage in total and for vaccines that should be done during childhood and adolescence. The error bars represent percentage along with 95% Confidence Interval.

**Table 1 children-09-01553-t001:** Sample characteristics for total sample (N = 501).

	N (%)
Gender	
Females	417 (85.8)
Males	69 (14.2)
Age, mean (SD)	22.2 (1.4)
Age	
18–22	244 (50.2)
23–30	242 (49.8)

**Table 2 children-09-01553-t002:** Full vaccination percentage in total sample and comparison according to demographics.

	Full Vaccination Percentage		P Student’s *t*-Test
	Mean %	95% CI	
Total sample	79	78–81	-
Gender			
Females	80	78–81	0.10
Males	76	72–81	
Age			
18–22	79	77–81	0.89
23–30	79	77–81	

**Table 3 children-09-01553-t003:** Full vaccination percentage regarding vaccines during childhood and during adolescence, in total sample and by their characteristics.

	Full Vaccination Percentage for Vaccines during Childhood		P Student’s *t*-Test	Full Vaccination Percentage for Vaccines during Adolescence		P Student’s *t*-Test
	Mean %	95% CI		Mean %	95% CI	
Total sample	79	78–80	-	78	75–81	-
Gender						
Females	80	78–81	0.045	78	74–81	0.48
Males	76	71–80		81	72–89	
Age						
18–22	79	77–81	0.76	78	73–82	0.83
23–30	79	77–81		78	74–83	

**Table 4 children-09-01553-t004:** Percentages of fully vaccinated participants in each vaccine, in total sample and by gender.

	Total Sample	Gender		
		Females	Males	
	N (%)	N (%)	N (%)	P
In adolescents				
Td or Tdap	238 (70.4)	201 (69.6)	37 (75.5)	0.40 +
Men ACWY	288 (100)	251 (100)	37 (100)	-
HPV ^1^	285 (75.6)	282 (76.6)	3 (33.3)	0.008 ++
In children < 10 years old				
DTaP/TdaP	243 (70.8)	205 (70)	38 (76)	0.39 +
Polio (IPV)	390 (83.3)	339 (84.1)	51 (78.5)	0.26 +
Hib	314 (70.6)	271 (70.6)	43 (70.5)	0.99 +
HBV	448 (95.1)	390 (96.3)	58 (87.9)	0.008 ++
PVC	45 (13.6)	39 (13.9)	6 (11.5)	0.64 +
MMR	463 (96.1)	400 (96.9)	63 (91.3)	0.04 ++
Var	404 (86.3)	354 (87.8)	50 (76.9)	0.017 +
HAV	372 (83)	321 (82.7)	51 (85)	0.66 +
MCC	406 (93.1)	350 (92.8)	56 (94.9)	0.78 ++
BCG	335 (73.8)	286 (73.3)	49 (76.6)	0.59 +
Mantoux *	121 (87.1)	108 (87.1)	13 (86.7)	>0.99 ++
Additional vaccines				
INFL	293 (64.1)	251 (63.9)	42 (65.6)	0.79 +
INFL **	91 (34)	78 (43.1)	8 (34.8)	0.45 +
MENB ^2^	43 (17.6)	37 (17.3)	6 (19.4)	0.78 +

* referred in those who had not been fully vaccinated for BCG ** in the last year (referred in those who had done it once), ^1^ recommended only for girls aged ≥ 11 years old, ^2^ recommended and reimbursed only for high-risk groups. + Pearson’s chi-square test ++ Fisher’s exact test. Note. The percentages presented are calculated without taking into account those who did not remember/ did not know if they had done the vaccine.

**Table 5 children-09-01553-t005:** Full vaccination coverage for health professionals (%) in total sample and by their characteristics.

	Full Vaccination Percentage for Health Professionals (%)		
	Mean	95% CI	P
Total sample	84%	82–85%	
Gender			
Females	83.90	82.4–85.5	0.31
Males	81.70	76.6–86.7	
Age			
18–22	84.10	81.8–86.3	0.54
23–30	83.10	81.1–85.2	

**Table 6 children-09-01553-t006:** Modifications of the Greek NIP since 1990.

Year	Modification
1991	Inclusion of the second dose of MMR for adolescents 11–12 years
1999	Shift of the second dose of MMR to children aged of 4–6 years
1998	Introduction of HBV vaccine
2006	Inclusion of MenC vaccine
2006	Introduction of the VAR vaccine (2 doses for children > 12 years old and 1 dose for younger children)
2008	Modification of the VAR vaccine to two doses for all children >15 months. Inclusion of the HAV vaccine (available since 1995). The Td booster was replaced by the Tdap. Recommendation of catch-up vaccinations for adolescents.
2011	Introduction of the MenACYW vaccine for adolescents aged of 11–16 years.

## Data Availability

The data presented in this study are available on request from the corresponding author.

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
