# Peer review of "Vaccination Coverage during Childhood and Adolescence among Undergraduate Health Science Students in Greece"

_children, 2022, doi:10.3390/children9101553_

Round 1

Reviewer 1 Report

The paper is well written and very interesting; it cover the aspect of the vaccination coverage during childhood and adolescence among undergraduate health science students in Greece.

Because of the available epidemiological data are based on sparse studies that provide only estimations of few samples, the aim of this work is to estimate complete vaccination rates of mandatory vaccinations according to the Greek NIP and evaluate the vaccination coverage of health sciences students of University of West Attica against VPDs recommended by the Greek Ministry of Health for HCWs. It could help universities health services to create programs to promote increased vaccination coverage among undergraduate students.

Comments:

The abstract includes a little introduction, some data related the Results section and the conclusion. I suggest you to edit the part related the results obtained ( because you will explain in the results section), explaining and justifying shortly the methods (use of statistical methods) and the data that you used in order to achieve your goals (without percentage related the results obtained).

On lines 135 you have inserted the section relating to Statistical Analysis, specifying the use of ANOVA. Why do you use ANOVA instead of other methods? The paragraph is very short and I recommend that you expand this section by explaining why you prefer using Anova for this type of data. Furthermore, it might be interesting to explain the input data used for this analysis.

In the Results section you have entered a different table but the caption labels are very short. I suggest you expand these labels to give readers a chance to clearly understand what the table represents. In particular I suggest expanding the caption labels of Table 1, Table 2, Figure 1 (where it might be interesting to explain how to read the plot). In figure 2 I suggest you to insert the label of the x-axis.

Line 194 lists various data in the discussion section. It may be more readable if you insert a table and / or chart.

Author Response

We appreciate the reviewers’ thoughtful reading of our work which undoubtedly improve our text. Detailed responses to each critique are provided below.

Please note that page and line numbers refer to the new marked copy of the manuscript that is attached.

 The paper is well written and very interesting; it cover the aspect of the vaccination coverage during childhood and adolescence among undergraduate health science students in Greece.

Because of the available epidemiological data are based on sparse studies that provide only estimations of few samples, the aim of this work is to estimate complete vaccination rates of mandatory vaccinations according to the Greek NIP and evaluate the vaccination coverage of health sciences students of University of West Attica against VPDs recommended by the Greek Ministry of Health for HCWs. It could help universities health services to create programs to promote increased vaccination coverage among undergraduate students.

Comments:

The abstract includes a little introduction, some data related the Results section and the conclusion. I suggest you to edit the part related the results obtained (because you will explain in the results section), explaining and justifying shortly the methods (use of statistical methods) and the data that you used in order to achieve your goals (without percentage related the results obtained).

Response: Thank you for your comment. According to reviewer’s suggestion we modified the Abstract section. Page 1 , lines 15-30.

On lines 135 you have inserted the section relating to Statistical Analysis, specifying the use of ANOVA. Why do you use ANOVA instead of other methods? The paragraph is very short and I recommend that you expand this section by explaining why you prefer using Anova for this type of data. Furthermore, it might be interesting to explain the input data used for this analysis.

Response: You are absolutely right.Thank you for your comment. It was our mistake to mention ANOVA. We have removed it from the revision. More details were added in the “Statistical Analysis ” section. Page 5 , lines 141-142.

In the Results section you have entered a different table but the caption labels are very short. I suggest you expand these labels to give readers a chance to clearly understand what the table represents. In particular I suggest expanding the caption labels of Table 1, Table 2, Figure 1 (where it might be interesting to explain how to read the plot). In figure 2 I suggest you to insert the label of the x-axis.

Response: Thank you for your comment. We have followed this suggestion. Clarification has been added to the caption labels of Table 1, Table 2, Figure 1 of the revised manuscript. In figure 2 the label of the x-axis was inserted.

Line 194 lists various data in the discussion section. It may be more readable if you insert a table and / or chart.

Response: We have followed this suggestion. This table has been added to the revised manuscript.

Reviewer 2 Report

After carful revision of the manuscript it was found that the manuscript covers an important topic and worthy of study and can be published after minor revision which does not effect the quality of the manuscript. I believe that some references could enrich this manuscript, for instance:

DOI 10.1093/ije/dyx052

and

DOI 10.1016/j.vaccine.2020.05.032

Author Response

We appreciate the reviewer's thoughtful reading of our work which undoubtedly improve our text. Detailed responses to each critique are provided below.

Please note that page and line numbers refer to the new marked copy of the manuscript that is attached.

After carful revision of the manuscript it was found that the manuscript covers an important topic and worthy of study and can be published after minor revision which does not effect the quality of the manuscript. I believe that some references could enrich this manuscript, for instance:

DOI 10.1093/ije/dyx052

and DOI 10.1016/j.vaccine.2020.05.032

Response: Thank you for your comment. We have followed this suggestion and the proposed references were added to the revised manuscript. Page 14 , lines 406-412.

Reviewer 3 Report

Thank you for giving me the opportunity to review your work. Overall, the manuscript is very well presented. The introduction gives a sufficient background about vaccinations in general with a specific focus on the requirements in Greece. Methods section is comprehensive and study design is appropriate. Results is clearly presented; tables and figures provide clear comparison between vaccination during childhood and puberty as well as correlation with demographics. Discussion is detailed and well-written with all limitations highlighted.   

Author Response

We appreciate the reviewer’s thoughtful reading of our work and we are grateful for the comments.